# Two New Flavonoids from the Nuts of *Areca catechu*

**DOI:** 10.3390/molecules24162862

**Published:** 2019-08-07

**Authors:** Mengfei Yuan, Yunlin Ao, Nan Yao, Jing Xie, Dongmei Zhang, Jian Zhang, Xiaoqi Zhang, Wencai Ye

**Affiliations:** Institute of Traditional Chinese Medicine & Natural Products, Guangdong Provincial Engineering Research Center for Modernazation of TCM, College of Pharmacy, Jinan University, Guangzhou 510632, China

**Keywords:** *Areca catechu*, flavonoids, absolute configuration, cytotoxicity

## Abstract

Two new flavonoids, calquiquelignan M (**1**), calquiquelignan N (**2**), along with nine known compounds (**3**–**11**), were isolated from the nuts of *Areca catechu* (Palmae). The new structures, including absolute configurations, were established by a combination of spectroscopic data and electronic circular dichroism (ECD) calculation. The known compounds were identified by comparing their spectroscopic data with reported in the literature. The flavonoids compounds (**1**–**8**) were evaluated for their cytotoxicity activities against three human cancer cell lines. Compounds **1** and **2** exhibited a moderate cytotoxic activity against HepG2 cell lines with IC_50_ values of 49.8 and 53.6 μM, respectively.

## 1. Introduction

Areca nut is obtained from the fruit of the *Areca catechu* (Palmae), which is widely distributed in southeast Asia and southern China [1]. Areca nuts are regarded as a traditional Chinese medicine usually used for the treatment of indigestion, liver disorders, and also used as a vermifuge [2,3]. Previous pharmacological studies of areca nuts have demonstrated antibacterial, antioxidant, anti-inflammatory, antifungal, and anthelmintic activities [4,5]. In a search for novel and bioactive constituents from Chinese herbal medicines [6,7,8,9,10], our group had reported three new areca alkaloids from the nuts of *A. catechu* [11]. In our continuing investigation, two new flavonoids calquiquelignan M (**1**), calquiquelignan N (**2**), and nine known compounds naringenin (**3**) [12], dihydrotricin (**4**) [13], sinesetin (**5**) [14], nobiletin (**6**) [15], 8-demethyleucalyptin (**7**) [16], eucalyptin (**8**) [16], (+)-isolariciresinol (**9**) [17], rhapontigenin (**10**) [18], glyceryl-2-vanillic acid methyl ester (**11**) [19], were isolated from the same plant parts as shown in Figure 1. The new structures, including absolute configurations, were elucidated by spectroscopic data and electronic circular dichroism (ECD) calculation, and the known ones were identified by comparison with data in the literature. Herein, we report the isolation, structure elucidation, and cytotoxic activities of these compounds.

## 2. Results and Discussion

### Structural Elucidation

Calquiquelignan M (**1**) was isolated as a yellow powder. The molecular formula of **1** was assigned as C_21_H_24_O_9_ based on its HRESIMS at *m*/*z* 443.1310 [M + Na]^+^ (calcd for C_21_H_24_O_9_Na, 443.1313), indicating 10 degrees of unsaturation. The IR absorption showed the characteristic absorptions for hydroxyl (3477 cm^−1^), carbonyl (1660 cm^−1^) groups, and aromatic ring (1535 and 1450 cm^−1^). The UV spectrum exhibited absorption maxima at 287 and 208 nm. The ^1^H-NMR spectrum indicated the presence of four aromatic protons [*δ*_H_ 6.71 (2H, s, H-2′/6′), 6.09 (1H, s, H-8), 6.08 (1H, s, H-6)]; two methines [δ_H_ 5.36 (1H, dd, *J* = 13.0, 3.0 Hz, H-2), 4.08 (1H, m, H-7′)]; three methylenes [δ_H_ 3.05 (1H, dd, *J* = 17.0, 13.0, H-3α), 2.82 (1H, dd, *J* = 17.0, 3.0, H-3β), 3.77 (2H, d, *J* = 3.4 Hz, H-8′), 3.85 (2H, d, *J* = 3.4 Hz, H-9′)], and three methoxyls [δ_H_ 3.91 (6H, s), 3.82 (3H, s)]. The ^1^^3^C-NMR spectrum displayed twenty-one carbon signals, including a carbonyl, eight quaternary carbons, six methines, three methylenes, and three methoxyls. With the aid of ^1^H–^1^H COSY, HSQC, and HMBC experiments, all of the ^1^H- and ^1^^3^C-NMR signals of **1** were assigned as shown in Table 1.

The above data of **1** resembled those of dihydrotricin (**3**) [12], except for the presence of one methoxy and one glycerin unit carbon signals. The ^1^H–^1^H COSY correlations between δ_H_ 3.85 (H_2_-8′) –4.08 (H-7′)–3.77 (H_2_-9′) confirmed the glycerin unit existence, and the HMBC correlations between H-7′ and C-4′ allowed the linkage between dihydroflavone fragment and glycerin fragment through the C-4′–O–C-7′ bond. The HMBC correlations between *δ*_H_ 3.82 (methoxy) and C-7 led to the increased methoxy substitution position at C-7 (Figure 2). Finally, the absolute configuration of **1** was deduced using the computational calculation method. The experimental ECD spectrum of **1** showed positive Cotton effects at 342 (Δ*ε* + 6.6), 260 (Δ*ε* + 29.3) nm and negative Cotton effects at 299 (Δ*ε* − 7.0), 216 (Δ*ε* − 18.2) nm, which were similar to those in the quantum chemical ECD calculation in Gaussian 09 software (Figure 3) [20]. Accordingly, the absolute configuration of **1** was determined as 2*S*.

Calquiquelignan N (**2**) was obtained as a yellow powder, possessed a molecular formula C_28_H_30_O_11_ as established by its HR-ESI-MS at *m*/*z* 565.1685 [M + Na]^+^ (calcd for C_28_H_30_O_11_Na, 565.1680). The IR absorption revealed the presence of hydroxyl (3443 cm^−1^), carbonyl (1651 cm^−1^) groups, as well as an aromatic ring (1511 and 1429 cm^−1^). The UV spectrum showed the characteristic absorption maxima at 287, 230, and 208 nm. The ^1^H-NMR spectrum indicated the presence of seven aromatic protons [δ_H_ 7.03 (1H, s, H-2′′), 6.89 (1H, d, *J* = 8.0 Hz, H-6′′), 6.87 (2H, s, H-2′/6′), 6.77 (1H, d, *J* = 8.0 Hz, H-5′′), 6.12 (1H, s, H-8), 6.08 (1H, s, H-6)]; three methines [δ_H_ 5.45 (1H, dd, *J* = 11.0, 6.0 Hz, H-2), 5.02 (1H, d, *J* = 6.7, Hz, H-8′), 4.14 (1H, m, H-7′)]; two methylenes [δ_H_ 3.18 (1H, dd, *J* = 16.5, 11.0, H-3*α*), 2.81 (1H, dd, *J* = 16.5, 6.0, H-3β), 3.79 (1H, m, H-9′a), 3.37 (1H, m, H-9′b)], and four methoxyls [δ_H_ 3.89 (6H, s), 3.85 (3H, s), 3.84 (3H, s)]. The ^1^^3^C-NMR spectrum displayed 28 carbon signals, including a chelated phenolic ketone carbon, eighteen olefinic carbons, three methines, two methylenes, and four methoxyls (Table 1). The above-NMR of **2** resembled those of **1**, except for the absence of a methylene, and the present of a 1′′,3′′,4′′-trisubstitution phenyl ring and a methine at (δ_C_ 74.5). The signal at δ_C_ 74.5 was assigned to C-8′ indicated the 1′′,3′′,4′′-trisubstitution phenyl ring substituted there, which was supported by the HMBC correlations between H-8′ and C-2′′/C-6′′. Furthermore, the HMBC correlations between H-7′ and C-4′ confirmed the C-4′–O–C-7′ bond existence, and correlations between methoxy (δ_H_ 3.84) and C-3′′, led to the methoxy substitution position at C-3′′ (Figure 4). Thus, the planar structure of **2** was determined, as shown in Figure 4.

Generally, adjacent protons of the *erythro* type have been reported to have smaller coupling constants (2.8–5.6 Hz) than those of the *threo* type (6.0–8.6 Hz) in different *d*-solvents [21,22,23]. Likewise, compound **2** was identified as a *threo-*configuration due to the 6.7 Hz coupling constants between H-7′ and H-8′ in CD_3_OD. Furthermore, the agreement of the ECD spectrum of **2** with those of **1** allowed the absolute configuration of **2** to be determined as 2*S* (Figure 3).

The pharmacological study of flavonoids and their derivatives showed that they possess cytotoxic activity [24]. Thus, the flavonoid compounds (**1**–**8**) were tested for their cytotoxic activity against three human cancer cells MCF-7, HepG2, and A-549 using MTT assay. Compounds **1** and **2** exhibited a moderate cytotoxic activity against HepG2 cell lines with IC_50_ values of 49.8 and 53.6 μM, respectively (Table 2).

## 3. Materials and Methods

### 3.1. General Experimental Procedures

Optical rotation was carried out using a Jasco P-1020 digital polarimeter (JASCO, Tokyo, Japan). UV spectra were obtained on a Jasco V-550 UV/VIS spectrometer (JASCO, Tokyo, Japan), and IR spectra on a Jasco FT/IR-480 plus infrared spectrometer (JASCO, Tokyo, Japan) with KBr discs. HR-ESI-MS data were detected on an Agilent 6210 ESI/TOF mass spectrometer (Agilent, Palo Alto, CA, USA). NMR spectra were recorded on Bruker AV-500 spectrometer (Bruker, Karlsruhe, Germany). Column chromatography (CC) was performed on silica gel (80~100 and 200~300 mesh; Qingdao Marine Chemical Inc., Qingdao, China), ODS (YMC, Kyoto, Japan) and Sephadex LH-20 (Pharmacia Biotech AB). Preparative HPLC was carried out on an Agilent 1260 system (G1311 pump and G1315D photodiode array detector) with a C_18_ reversed-phase column (20 × 250 mm, 5 μm, Shiseido Fine Chemicals Ltd., Osaka, Japan).

### 3.2. Plant Material

The areca nuts were collected in Sanya, Hainan province, P. R. China, in October 2014, and identified by Professor Guang-Xiong Zhou (College of Pharmacy, Jinan University). A voucher specimen (No. CP2014101403) was deposited at the herbarium of the College of Pharmacy, Jinan University, Guangzhou, P. R. China.

### 3.3. Extraction and Isolation

The air-dried and powdered areca nuts (30.0 kg) were extracted at room temperature with 95% EtOH to afford a residue (1.1 kg), which was then suspended in H_2_O and treated with 0.5% hydrochloric acid to adjust the pH to 2 to 3. After extraction with CHCl_3_, the CHCl_3_ extract (60.0 g) was subjected to silica gel column chromatography and eluted with CHCl_3_–CH_3_OH (100:0 → 0:100) to afford 8 fractions (Fr. A–H). Fr. D (2.8 g) was chromatographed on Sephadex LH-20 (CHCl_3_–CH_3_OH, 1:1) and HPLC (CH_3_OH–H_2_O, 35:65) to afford **1** (7.2 mg, *t*_R_ 14.3 min), **2** (6.8 mg, *t*_R_ 28.7 min), **3** (10.4 mg, *t*_R_ 37.6 min). Fr. E (5.9 g) was separated by Sephadex LH-20 (CHCl_3_–CH_3_OH, 1:1) and further HPLC (CH_3_OH–H_2_O, 50:50) to afford **9** (35.8 mg, *t*_R_ 17.8 min) and **11** (26.3 mg, *t*_R_ 29.8 min). Fr. F (2.3 g) was subjected to ODS CC (CH_3_OH–H_2_O, 1:9 → 7:3) and HPLC (CH_3_OH–H_2_O, 40:60) to yield **4** (9.2 mg, *t*_R_ 40.4 min), **7** (6.7 mg, *t*_R_ 25.8 min), **8** (8.3 g, *t*_R_ 33.8 min), **10** (12.3 g, *t*_R_ 19.6 min). Fr. G (1.9 g) was separated by Sephadex LH-20 (CHCl_3_–CH_3_OH, 1:1) and further HPLC (CH_3_OH–H_2_O, 40:60) to afford **5** (12.2 mg, *t*_R_ 24.1 min) and **6** (10.5 mg, *t*_R_ 35.8 min).

Calquiquelignan M (**1**): yellow powder; ${\left[\alpha \right]}_{D}^{25}$ − 8.9 (*c* 0.36, CH_3_OH); UV (CH_3_OH) λ_max_ (log *ε*): 208 (4.14), 287 (3.88) nm; CD (CH_3_OH, Δ*ε*) *λ*_max_: 342 (+6.6), 299 (−7.0), 260 (+29.3), 216 (−18.2) nm; IR (KBr) *ν*_max_ 3477, 1660, 1621, 1535, 1450, 1304, 1158, 1114 cm^−1^; ^1^H- and ^1^^3^C-NMR spectral data, see Table 1; HR-ESI-MS: *m*/*z* [M + Na]^+^, calcd for C_21_H_24_O_9_Na: 443.1313, found: 443.1310.

Calquiquelignan N (**2**): yellow powder; ${\left[\alpha \right]}_{D}^{25}$ − 13.5 (*c* 0.47, CH_3_OH); UV (CH_3_OH) λ_max_ (log *ε*): 208 (4.45), 230 (4.02), 287 (4.05) nm; CD (CH_3_OH, Δ*ε*) *λ*_max_: 324 (+2.5), 299 (−7.9), 250 (+3.4), 211 (−11.3) nm; IR (KBr) *ν*_max_ 3443, 1651, 1511, 1429, 1362, 1032, 826 cm^−1^; ^1^H- and ^1^^3^C-NMR spectral data, see Table 1; HR-ESI-MS: *m*/*z* [M + Na]^+^ calcd for C_28_H_30_O_11_Na: 565.1680, found: 565.1685.

^1^H- and ^13^C-NMR spectra of these compounds are available in the Appendix A.

### 3.4. Computational Calculation

Conformational searches were performed in the Sybyl 8.1 software by using the MMFF94S molecular force field, which afforded 12 conformers for **1**, with an energy cutoff of 10 kcal/mol. The ECD calculation for the optimized conformers was carried out using time-dependent DFT (TDDFT) methods at the B3LYP/6-31+G(d) level in the gas phase by using Gaussian 09 software. The overall ECD curves of **1** were weighted by Boltzmann distribution of each conformer (with a half-bandwidth of 0.3 eV). The calculated ECD spectra of **1** were subsequently compared with the experimental ones. The ECD curves were produced by SpecDis 1.6 software (University of Wuerzburg, Bavaria, Germany).

### 3.5. Cytotoxicity Assay

The MTT assay for the determination of the cytotoxicity was performed as described previously [25]. Briefly, cancer cells were plated into 96-well plates. After 48 h of preculture, the cells were treated with compounds at various concentrations for 72 h and then stained with MTT. Absorbance at 570 nm was measured on a microplate reader.

## 4. Conclusions

In summary, we isolated two new flavonoids, calquiquelignan M and N (**1**–**2**) and nine known compounds naringenin (**3**), dihydrotricin (**4**), sinesetin (**5**), nobiletin (**6**), 8-demethyleucalyptin (**7**), eucalyptin (**8**), (+)-isolariciresinol (**9**), rhapontigenin (**10**), glyceryl-2-vanillic acid methyl ester (**11**) from the nuts of *A**. catechu*. The new compounds were elucidated by spectroscopic analyses, and computational calculation, and the known ones were identified by comparing their spectroscopic data with reported in the literature. Moreover, all of the flavonoids (**1**–**8**) were evaluated for their cytotoxicity activities against three human cancer cell lines. Compounds **1** and **2** showed moderate cytotoxicity against HepG2 cell lines with IC_50_ values of 49.8 and 53.6 μM, respectively.

## Figures and Tables

**Figure 1 molecules-24-02862-f001:**
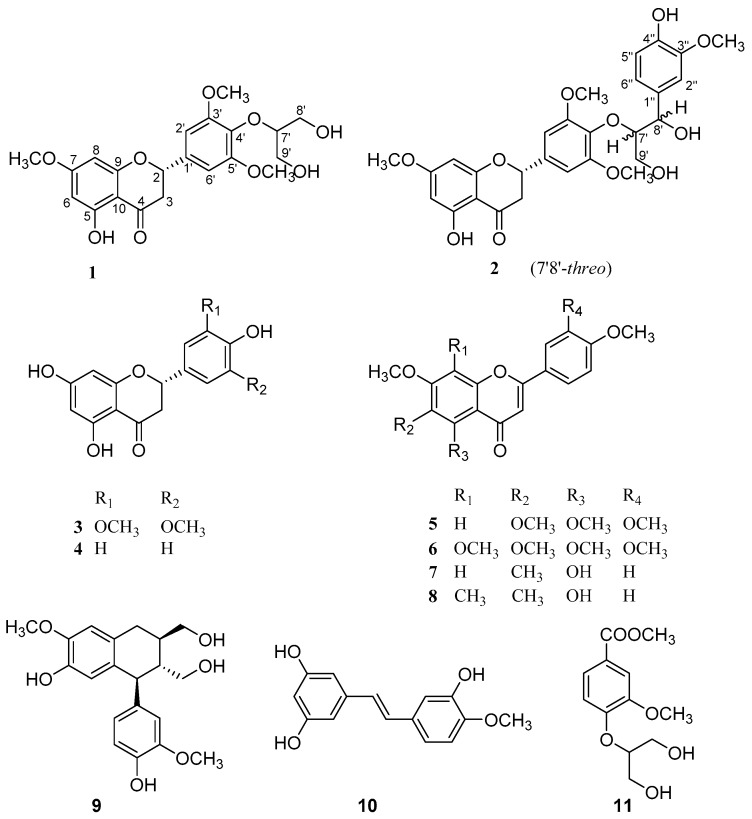
Chemical structures of **1** to **11**.

**Figure 2 molecules-24-02862-f002:**
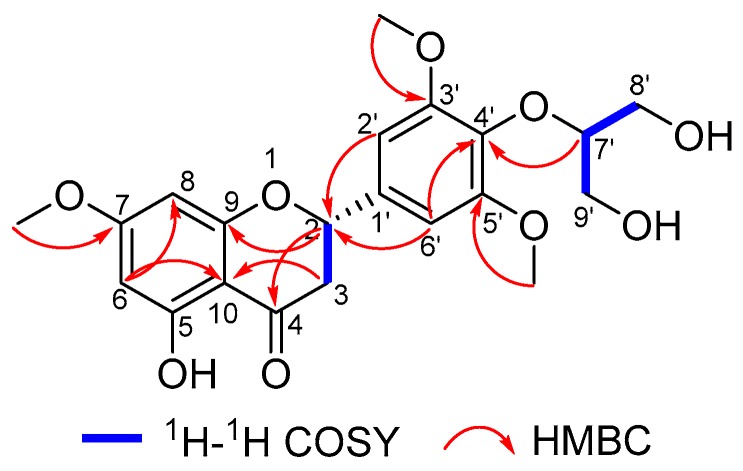
^1^H–^1^H COSY and key HMBC correlations of **1.**

**Figure 3 molecules-24-02862-f003:**
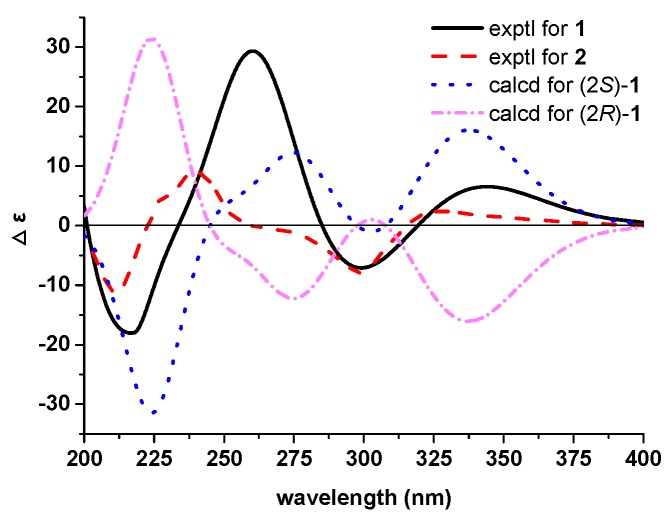
Experimental electronic circular dichroism (ECD) spectra for **1** and **2**; and calculated for **1**.

**Figure 4 molecules-24-02862-f004:**
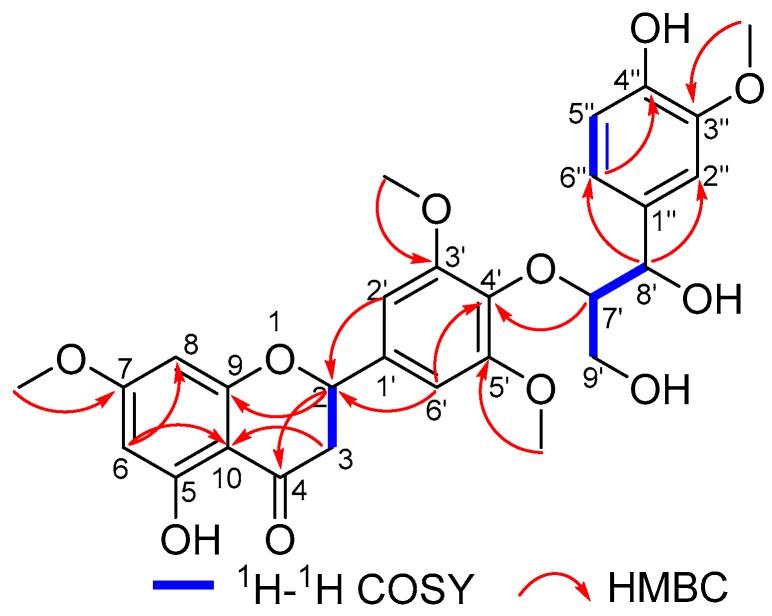
^1^H–^1^H COSY and key HMBC correlations of **2**.

**Table 1 molecules-24-02862-t001:** ^1^H- and ^1^^3^C-NMR data of **1** and **2** (δ in ppm, *J* in Hz).

Position	1 ^a^				2 ^b^	
		δ_H_	δ_C_		δ_H_	δ_C_
2		5.36 dd (13.0, 3.0)	79.4		5.45 dd (11.0, 6.0)	80.7
3	α β	3.05 dd (17.0, 13.0) 2.82 dd (17.0, 3.0)	43.8		3.18 dd (16.5, 11.0) 2.81 dd (16.5, 6.0)	44.4
4		-	195.5		-	197.8
5		-	164.4		-	165.3
6		6.08 s	95.4		6.08 s	95.9
7		-	168.2		-	169.6
8		6.09 s	94.6		6.12 s	95.1
9		-	162.6		-	164.4
10		-	103.2		-	104.1
1′		-	135.0		-	136.5
2′/6′		6.71 s	103.4		6.87 s	104.9
3′/5′		-	153.6		-	154.4
4′		-	135.7		-	137.4
7′		4.08 m	85.0		4.14 m	88.9
8′		3.85 d (3.4)	62.7		5.02 d (6.7)	74.5
9′	α b	3.77 d (3.4)	62.7		3.79 m 3.37 m	61.8
1′′		-	-		-	133.5
2′′		-	-		7.03 s	111.7
3′′		-	-		-	148.7
4′′		-	-		-	147.2
5′′		-	-		6.77 d (8.0)	115.8
6′′			-		6.89 d (8.0)	120.8
7-OCH_3_		3.82 s	55.9		3.85 s	56.4
3′-OCH_3_		3.91 s	56.5		3.89 s	56.8
5′-OCH_3_		3.91 s	56.5		3.89 s	56.8
3′′-OCH_3_		-	-		3.84 s	56.3

^a^ Measured in CDCl_3_. ^b^ Measured in CD_3_OD.

**Table 2 molecules-24-02862-t002:** Cytotoxicity of compounds **1**–**8** (IC_50_, μM).

Compound	MCF-7	A-549	HepG2
**1**	>100	>100	49.8
**2**	>100	>100	53.6
**3**	>100	>100	>100
**4**	>100	>100	>100
**5**	>100	>100	>100
**6**	>100	>100	89.6
**7**	>100	>100	>100
**8**	>100	>100	>100
cisplatin	19.8	15.3	17.6

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
