# Peer review of "Two New Flavonoids from the Nuts of Areca catechu"

_molecules, 2019, doi:10.3390/molecules24162862_

Round 1
Reviewer 1 Report
The paper describes two new flavonoids and nine known compounds identified from Areca nuts. The results are clearly showed and should be accepted for publication.
Only a unique suggestion :
Authors should insert the family name in the introduction as well as in the abstract.
Author Response
Answer: Thank you very much for your suggestion. We have added the family name in the introduction as well as in the abstract in present manuscript.
Reviewer 2 Report
The manuscript written by Yuan et al. describes the identification of two new flavonoids, calquiquelignan M and calquiquelignan N, from the nuts of Areca catechu. Their manuscript is concise and well organized. The authors evaluated cytotoxicity of the compounds isolated from the plant; however, there are no data in the manuscript. The authors are required to add a table about the cytotoxicity data of the compounds to MCF-7 cells, HepG2 cells, and A-549 cells. The authors are also required to compare their cytotoxicity data with previous papers reporting the cytotoxicity of the known compounds (compound 3 to 8).
Minor point
Line149: Replace "exposure" with "preculture".
Author Response
Answer: Thank you very much for your comment and suggestion. We have added the table about the cytotoxicity data of the compounds to MCF-7 cells, HepG2 cells, and A-549 cells in the manuscript. By searching the literature, we found there are no reported about the cytotoxicity of compounds 4, 5, 7, and 8, and compound 6 showed certain cytotoxicity against HepG2 cells.
Answer: Line 149 (now is Line 152), "exposure" was revised as "preculture" in present manuscript.
Reviewer 3 Report
The manuscript is clear, well organized, and reports some interesting results. However, the authors should insert a section discussing the results obtained. In addition, they should include a table of the toxicological test results comparing those obtained at the different concentrations tested in the various cell lines. Finally, the manuscript’s English grammar and style should be revised by a native speaker.
Author Response
Answer: Thank you very much for your comment and suggestion. We have added the Conclusions section and the table about the cytotoxicity data of the compounds to MCF-7 cells, HepG2 cells, and A-549 cells in the manuscript. The present manuscript also has been revised by a native speaker.